# Neuromodulation for Craniofacial Pain and Headaches

**DOI:** 10.3390/biomedicines11123328

**Published:** 2023-12-16

**Authors:** Ray J. Pak, Jun B. Ku, Alaa Abd-Elsayed

**Affiliations:** 1Department of Physical Medicine and Rehabilitation, New York Medical College, Metropolitan Hospital, New York, NY 10029, USA; kuj@nychhc.org; 2Department of Anesthesia, University of Wisconsin, Madison, WI 53792, USA

**Keywords:** neuromodulation, headache, facial pain, stimulation, neuralgia, migraine, cluster

## Abstract

Headaches and facial pain are highly prevalent diseases but are often difficult to treat. Though there have been significant advances in medical management, many continue to suffer from refractory pain. Neuromodulation has been gaining interest for its therapeutic purposes in many chronic pain conditions, including headaches and facial pain. There are many potential targets of neuromodulation for headache and facial pain, and some have more robust evidence in favor of their use than others. Despite the need for more high-quality research, the available evidence for the use of neuromodulation in treating headaches and facial pain is promising. Considering the suffering that afflicts patients with intractable headache, neuromodulation may be an appropriate tool to improve not only pain but also disability and quality of life.

## 1. Introduction

Headaches are a common complaint, with an estimated lifetime prevalence of 95% and a general prevalence of 48.9% [1]. They are classified based on their characteristics as outlined in the International Classification of Headache Disorders (ICHD); primary headaches are classified as migraine, tension-type, trigeminal autonomic cephalgia (e.g., cluster headache), or other primary headache disorders. Secondary headaches or other facial pain have a larger differential, including trauma to the head and/or neck, cranial or cervical vascular disease, nonvascular intracranial disorders, substance use or withdrawal, infection, a disorder of homeostasis, other structural disorder, psychiatric disorders, or other. As one might expect from the terminology, primary headaches have no known underlying cause, while secondary headaches result from another condition that may cause traction or inflammation of pain-generating structures, such as the trigeminal nerve [2]. Primary headaches constitute nearly 98% of all headaches, but secondary headaches are important to recognize as they may often be a consequence of life-threatening disorders [1].

Facial pain may be classified according to reported symptoms and history. The differential diagnosis is broad, including pain of musculoskeletal origin, dental pain, primary headache, neuralgia, neuropathy, etc. [3,4,5,6,7].

Though most headaches may not be life-threatening, they are still a source of significant discomfort for patients, and so treatment is an important consideration for physicians. Primary headache management varies depending on type but generally includes some type of abortive medication, as well as preventive if frequency is high enough. For example, in the case of migraines, a patient may use acetaminophen, an NSAID, or triptan (or some combination thereof) as an abortive and take a daily amitriptyline for prevention. There are a few procedures that are also used for headaches. OnabotulinumtoxinA, commonly known as “Botox” and nerve blocks (specifically of the greater occipital nerve) are also treatment options for migraine prophylaxis [1,2].

Neuromodulation has been gaining significant interest and popularity in the treatment of various chronic pain disorders, including neuropathic and back pain. It refers to a technique that uses pulsed electrical energy near a nerve or spinal cord, using leads implanted into a nearby space. This technique is based on the gate control theory of pain, initially proposed by Melzac and Wall in 1965. The conventional understanding of how neuromodulation works to reduce pain is that by stimulating larger A-beta fibers, pain signals carried by smaller C- and A-delta fibers may be interrupted [8,9]. It is stated that the first reported clinical application of spinal cord stimulation was two years later, but its popularity has been increasing significantly since [10]. The first reported case in the management of intractable headaches was in the late 1990s [11]. This paper explores the current literature available on PubMed on the topic of neuromodulation in the treatment of headache and facial pain.

## 2. Literature Review

### 2.1. Occipital Nerves

The occipital nerves are some of the most well-studied for their therapeutic potential in the treatment of headaches. They are composed of three nerves originating from the C2 and C3 spinal nerves. They are responsible for the innervation of the posterior scalp and lateral scalp behind the ear, as well as possible contributions to the facet between C2 and C3 spinal nerves [12]. Stimulation of the occipital nerves has been studied for the treatment of a variety of head and facial pain pathologies, including neuralgia, cluster, and migraine headaches.

#### 2.1.1. Occipital Neuralgia

The first reported use of occipital nerve stimulation (ONS) was in 1999 by Weiner and Reed in the treatment of occipital neuralgia (ON). They recruited thirteen patients with intractable neuralgia. One patient was able to explant the device three years after initial implantation due to the resolution of neuralgia. Follow-up for the other twelve of the patients was a mean of over 2 (range 1–6) years. Here, 67% reported greater than 75% pain relief and 33% reported greater than 50% pain relief [13].

One case report demonstrated a profound effect of ONS in terms of reducing pain and the resumption of normal activity in a patient with ON. The baseline pain was scored as being 6/10. She started with temporary and then permanent placement due to pleasant results. She scored 0/10 since the permanent ONS started, and this persisted through five- and twelve-month follow-ups [14].

Another three patients were implanted with ONS for ON. The average follow-up period was nine (range 6–16) months, and there was a reported 55% (range 25–90%) pain relief [15].

Ten patients with ON and ten with transformed migraine (discussed below) underwent subcutaneous placement of a C1-2-3 paddle-style electrode for ONS after a two-week trial, with at least a 50% reduction in pain. Symptom duration ranged from eight months to ten years. At one-month follow-up, 80% of patients reported a greater than 90% pain reduction and 20% reported a 75–90% reduction. Of nine patients who had completed a six-month follow-up, 78% reported a greater than 90% reduction, 11% a 50–75% reduction, and another 1% with a less than 50% reduction. However, 95% of patients reported improvement in quality of life. One patient with three years of successful stimulation had loss of pain control, which was reversed after interrogation revealed battery depletion [16].

Six patients with ON for an average of 4.9 years were treated with ONS after a seven- to fifteen-day trial. At three-month follow-up, pain, as measured through the use of the visual analog score (VAS), decreased by 71%, and the pain disability index scores decreased by 72%, which were statistically significant [17].

In a retrospective analysis, fourteen patients (ten permanently implanted) with intractable ON were treated with ONS after a successful 5–7-day trial. The mean follow-up was 22 (range 5–32 months). At final follow-up, 50% of the patients were pain-free, and 30% reported significant relief. Here, 10% had the stimulator removed due to the resolution of pain that did not return after explantation [18].

A retrospective review of 60 patients treated with ONS for occipital headaches after a positive response to the TENS trial had seen a statistically significant 72.2% decrease in mean pain score (as measured by VAS) as well as a 72.7% reduction in number of headache days per month after one year. In addition, 13% of patients had a mean VAS of 0 or 1/10. Furthermore, 43% of patients only used acetaminophen as needed and were able to cease the use of other pain medications. Researchers report follow-up between 13 and 72 months with stable results [19].

#### 2.1.2. SUNCT/SUNHA

Here, 31 patients with short-lasting unilateral neuralgiform headache (SUNCT syndrome) were also trialed on bilateral ONS in an uncontrolled, open-label prospective study. At final follow-up for patients (range 13–89 months with a mean of 44.9 months), the mean daily attack frequency had reduced by a statistically significant 69%. In this study, 38.7% of patients were pain-free at final follow-up, with a mean complete remission time of 36.5 months [20].

#### 2.1.3. Hemicrania Continua

In a crossover study, six patients with hemicrania continua were implanted with a suboccipital nerve stimulator ipsilateral to their chronic headache. After a median follow-up of 13.5 months (range 6–21 months), 67% of patients reported an 80–95% improvement, 17% reported a 30% improvement, and 17% reported a worsening of pain by 20%. After the device was switched off, headaches did not recur for days to weeks for most patients [21].

#### 2.1.4. Trigeminal Neuralgia

A retrospective study examined the effects of ONS on resistant trigeminal neuralgia (TN) without painful trigeminal neuropathy. Seven patients with refractory TN who had received ONS were included. Pain was measured through the application of the Barrow Neurological Institute (BNI) pain score and mean pain relief at best and at last follow-up. There was an improvement in BNI score after ONS (from BNI V, defined as severe pain with no relief, to BNI IIIa, defined as no pain with continued mediation, at best and at last follow-up). The mean follow-up after implantation was 59 (range, 2–106; median, 73) months. At best, mean pain relief was 86.7% (range, 70–100%; median, 85%) and at last follow-up, it was 58.0% (range, 0–100%; median, 50%). No one was able to cease all medication, but some were able to decrease their medication intake [22].

#### 2.1.5. Cluster Headache

A prospective, uncontrolled study of ten patients who had ONS for intractable chronic cluster headache (cCH) was conducted. Pain scores (numeric rating scale (NRS)) and Short Form 36 questionnaires for the assessment of quality of life (SF-36) were obtained at follow-up. Patients were followed up at one and three months, then every three months thereafter. Here, 90% of patients had improvement in cluster headaches when treated with ONS, with a mean overall improvement of 44% (range, 20–90%) in terms of attacks. Daily frequency also dropped from a mean of six (2–14) to three (range 0.4–11) and intensity from a mean baseline of 8 (range, 6–9) to 6 (range, 2–9). Here, 30% of patients also had up to a nineteen-day pain-free period. In this study, 70% of patients reduced acute medication by up to 69% (range, 25–100%), and 30% of patients started a dose decrease in preventive medication. All patients had improvements in SF-36, but the results were not statistically significant [23].

In a prospective pilot study on the treatment of cCH, eight patients (mean age 45.3 years) had a suboccipital neurostimulator implanted. The mean duration of symptoms was 13.6 years. The mean follow-up was 15.1 (range 3–22) months. Here, 25% of patients were pain-free at 16 and 22 months, 38% had an approximately 90% reduction in attack frequency, while 25% experienced an improvement of approximately 40%. Many were able to reduce and one was able to stop preventive medication [24]. The authors conducted a larger prospective study of fifteen patients (with the eight patients cited previously) who received suboccipital stimulators on the side of their cluster headaches. The mean follow-up period was 36.82 months, and nearly 80% of patients had at least a 90% improvement in pain; 60% were pain-free at longer-term follow-up. About 13% of patients had no or mild improvement [25]. 

A larger monocentric, open-label study including 35 cCH patients who had undergone ONS had a median follow-up of 6.1 (range 1.6–10.7) years. The mean number of daily attacks dropped by a statistically significant 57.9%, from 5.7 to 2.4. In addition, 66.7% of the patients in the PP analysis had at least a 50% reduction in the number of headaches per day. In this study, 33.3% of patients were non-responders, though 50% of these did initially have at least a 50% reduction in the number of headaches per day, lasting a mean of 14.6 (2–48) months. Though no patients were able to stop CH prophylaxis, 66.7% were able to stop long-term steroids, which were used to try to control the headaches [26].

Fourteen patients with intractable cCH were implanted with bilateral ONS. The median follow-up for the patients was 17.5 (range 4–35) months and 71% had demonstrated improvement in frequency, severity, or duration. The effect on frequency was most pronounced. Though none had full remission of pain, 30% had at least a 90% improvement, 30% at least 40%, and 40% had a 20–30% improvement [27]. A year prior, the authors published a report on eight patients (median age 46 (range 32–57) years old) with intractable cCH treated with bilateral ONS. Symptom duration was a median of six (range two to twelve) years, and median follow-up was 20 (range 8–27) months. In this study, 75% of the patients reported improvement in their condition, but none were pain-free with treatment. Furthermore, 25% of patients had at least a 90% improvement in attacks, 38% had at least a 40% improvement, and 13% had at least a 25% improvement. Moreover, 13% were able to stop abortive medication completely, while 38% were able to reduce but not eliminate their medication [28].

In a cohort study of thirteen patients with intractable cCH implanted with bilateral ONS, researchers found that at a mean 14.6-month follow-up duration (range 3–34 months), 77% of patients experienced significant improvement within a few days of implantation. Furthermore, 15% had delayed but maintained improvements, while less than 1% had no improvement at all. Eight patients achieved a 1-year follow-up with a mean attack frequency decrease of 68%, and mean attack intensity in terms of NRS decreased by 49% by last visit. Interestingly, two patients who had follow-up after one year with active seasonal stimulation were found to have recurrent bouts of cluster headaches, while one became completely refractory to ONS after sixteen months [29].

In another retrospective case series of seventeen patients with either cCH (five) or CM (twelve) who had received burst ONS, researchers found that 60% of the cCH had complete remission and 100% had had at least an 80% reduction in attack frequency, with a mean reduction of 42% in residual headache intensity. The migraine group showed a variable response; 33% had a reduction in both headache days and intensity, 33% had a reduction in headache days but not intensity, 17% had a reduction in headache intensity but not days, and 17% had no improvement in either intensity or days [30].

In this study, 51 patients with cCH (nineteen with additional CM, three with additional short-lasting unilateral neuralgiform headache attacks (SUNHAs), and three with additional CM and SUNHAs) were implanted with bilateral ONS without trial stimulation. The mean follow-up time was 39.17 months, with four patients having their implants removed by the time of follow-up. At follow-up of the whole cohort, 52.9% of patients had at least a 50% reduction in daily attack frequency (responder rate), and 47.1% of patients reported over six months of continuous freedom from pain (the mean duration of pain freedom was 16.25 months, with a range of 6–48 months). Significant improvements were also noted in attack intensity, duration, disability (as measured through the use of migraine disability assessment (MIDAS)), headache impact on living (as measured via the Headache Impact Test (HIT-6)), and depression (as measured using the Beck Depression Inventory II (BDI-II)). There were also significant decreases in Hospital Anxiety and Hospital Depression Scores—Anxiety Component and Depression Component (HAD-A and HAD-D, respectively). Quality of life, as measured in relation to the Euro-QoL 5D index (EQ5D), Euro QoL visual analog score (EQ-VAS), and SF-36 P (physical component) and SF-36 M (mental component) was not significantly improved, except for SF-36 M. In the 32 patients with cCH alone, the mean follow-up time was 42.59 (range 2–81) months with a 53.1% responder rate, also with significant improvements in daily attack severity and duration, MIDAS, HIT-6, HAD-A, HAD-D, and BDI-II. There were no significant improvements in quality-of-life metrics. For the nineteen patients with multiple headache phenotypes, the mean follow-up time was 33.42 (range 13–76) months, with a responder rate of 52.6% and significant improvements in attack intensity and duration. This group also had significant improvements in HIT-6 and EQ-VAS but in no other disability, affect, or quality-of-life metric. Out of twenty-seven patients taking preventive medications at baseline, four were able to cease all preventive medication use, and seventeen were able to make reductions [31].

A prospective, multicenter observational study was conducted to study 67 patients who had bilateral ONS for intractable cCH (52 and 44 by the time of 3-month and 12-month follow-up, respectively). This study was particularly interested in the functional and emotional impacts of headache on quality of life, in addition to efficacy in terms of pain relief. Patients were broadly classified as excellent, mild, and non-responders based on the patient’s global impression of change (PGIC), reduction in attack frequency, and prophylactic treatment changes. At three months, 75% of patients experienced at least a 30% reduction in attack frequency, while at twelve months, 64% had at least a 30% reduction in attack frequency, and 59% had at least a 50% reduction. Prophylactic treatment could be decreased in 52% of patients at three months, which decreased to 40% of patients by twelve. By twelve months, 22% of patients were considered non-responders. Of this group, 67% had not experienced any degree of improvement with ONS at all, while 33% had some improvement earlier but appeared to have not been optimally stimulated for technical reasons (e.g., battery depletion, hardware dysfunction, etc.) at the time of follow-up. Other metrics, including mean attack frequency, intensity, and duration, as well as functional and emotional impacts, were significantly improved after ONS. EQ-5D VAS significantly improved in excellent responders while improving but not reaching significance in mild responders [32].

A multicenter, randomized, double-blind, phase 3, electrical dose-controlled clinical trial was conducted with a twelve-week baseline observation period, followed by a ten-day ONS run-in phase as well as a 24-week randomized, double-blind ONS treatment period with a stepwise increase in ONS intensity and a 24-week open-label ONS treatment phase. The primary outcome was the mean attack frequency (MAF) per week in the last four weeks of the masked study period. In this study, 131 patients with intractable cCH (mean duration four years; mean age 44 years) were randomly allocated to receive either 100% ONS or 30% ONS. There was a significant decrease in median weekly MAF after ONS onset by 5.21 from 7.38 and no significant difference between groups between weeks 21 and 24, with not much more improvement in the open-label phase. Between weeks 21–24 and 45–48, about half of the total study participants had at least 50% pain relief. In this study, 5% were pain-free between weeks 1 and 4, 7% between weeks 21 and 24, and 12% between weeks 45 and 48 [33]. 

#### 2.1.6. Migraine

In an uncontrolled case series, twenty-five patients with intractable episodic migraine (EM) (suggestive criteria for transformed migraine) for a median of 10 (range 1–30) years were treated with ONS. The average length of follow-up was 18.3 (range 9–36) months). Mean headache frequency was reduced from 75.6 days at baseline to 37.5 after treatment, which was statistically significant. In addition, on a NRS of 0–10, the mean severity decreased from 9.32 to 5.72, which was also statistically significant. In this study, 88% of patients had at least a 50% reduction in headache frequency or severity. Disability, as measured via MIDAS, decreased from 121 (severe) to 15 (disability) [34].

Ten patients with ON (discussed above) and ten with transformed migraine underwent subcutaneous placement of a C1-2-3 paddle-style electrode for ONS after a two-week trial with at least a 50% reduction in pain. Symptom duration ranged from 2 to 25 years. At one-month follow-up, 90% of patients reported a greater than 90% pain reduction, and 10% reported a 75–90% reduction. Of nine patients who had completed a six-month follow-up, 78% reported a greater than 90% reduction, and 22% reported a 75%-90% reduction. There was a reported near-total resolution of migraine disability and medication requirement (90.1%) as well [16].

In one prospective, long-term, open-label, uncontrolled observational study, 41 patients (4 excluded) with various locations of facial pain or headache were enrolled to receive ONS. The average age was 46.9 (range 27–74) years, and the follow-up time was 9.4 ± 6.1 years; 31 of the patients completed a seven-year follow-up period. In this study, 36 patients had favorable responses to a trial and progressed to permanent implantation. Most patients had a significant reduction in pain intensity, with VAS decreasing by 4.9, and frequency, with the number of migraine days per month decreasing by 59.2%, which remained stable through the entire follow-up. Furthermore, 13.9% of those with permanent implantation were pain-free at their last visits. They also noted improved social activities and sleep quality. Patients also reduced their average oral medication use from a baseline of 4.4 to 1.3 at last follow-up, with 40% not using any analgesic medications [35]. 

A multicenter, uncontrolled study was conducted with 112 (108 at 3-month visit, 97 at 9-month visit, and 91 at 12-month visit) patients treated with ONS for the treatment of intractable chronic migraine (CM). At 3-month follow-up, the mean reported pain relief was 25.1%, with 21.7% of patients reporting at least 50% pain relief. At 9- and 12-month follow-ups, the reported relief was 30.8%, with 28.5% reporting at least 50% relief. At 3-month follow-up, there was an average 16% decrease in the number of headache days (70.2 to 55) [36].

A retrospective analysis of fifteen patients who had implanted ONS (eight bilateral and seven unilateral) after a trial for the treatment of medically intractable headache was conducted (eight CM, three cCH, two post-traumatic, and two hemicrania continua). The researchers measured headache frequency, severity, disability (MIDAS), headache impact on living (HIT-6), and depression score (as measured via BDI-II). The mean follow-up was 17.8 (range 6–36) months. There were statistically significant improvements noted in headache frequency in the past ninety days (from ninety to sixty days), severity (from 6.75/10 to 4.5/10), MIDAS (from 170 to 116), HIT-6 (from 73 to 61), and BDI-II (from 19 to 11.5) as compared to baseline. Here, 60% had at least a 30% reduction in headache severity and/or frequency, with 53% with at least a 50% reduction [37].

In a case series, eight patients (mean 44 years old) with CM were implanted with bilateral suboccipital stimulators. Symptom duration ranged from eight months to approximately thirty years (not specifically stated in the paper). The mean follow-up was 1.5 years (range 7 months to 3 years). In this study, 50% reported excellent responses, being pain-free with only rare breakthrough headaches. Furthermore, 25% reported very good responses with suppression of headaches most of the time but about 10 days per month of breakthrough headaches for which they may increase stimulation or take an analgesic. The other 25% reported improvements (reduced severity of headache by 50–75%) but continued to have constant headaches. All patients were able to considerably reduce or completely stop their headache medications [38].

Moreover, 34 patients (29 completing the study) were implanted with ONS for the treatment of CM, with 85% of patients also meeting the criteria for medication-overuse headache (MOH) in a prospective, randomized cross-over study. The two arms of the study were stimulation on and stimulation off. All patients underwent a trial period between 15 and 30 days, and those who did not respond were explanted. Here, 32 patients were assessed at an average 45 ± 23 days (range: 12–122 days) after lead implantation during the stimulation period. Of these, 9% reported at least a 50% reduction in the number of attacks, 22% reported a reduction of at least 50% in the severity of attacks, 66% reported a reduction of at least 50% in both number and severity, with only 3% not achieving at least a 50% reduction in number or severity of attacks. Follow-up was carried out at one, three, six, and twelve months from implantation. Compared to baseline, every follow-up visit, MIDAS and SF-36 scores were significantly improved. There was also a significant decrease in the median monthly dose of medication use [39].

In a randomized, double-blind controlled trial (RCT) of 268 subjects with CM who were implanted with ONS (157 implanted with a permanent implant) or control for twelve weeks followed by open-label treatment for forty more weeks. The researchers analyzed a separate group who met the criteria for intractable CM (ICM) in addition to all patients (ITT). In this study, 59.5% of patients reported achieving a 30% reduction in headache days and/or pain intensity (VAS), and 47.8% reported achieving a 50% reduction. In addition, MIDAS scores were significantly reduced for both groups [40]. 

Another RCT, known as the “ONSTIM” feasibility study, of 75 subjects receiving ONS for intractable CM separated the treatment group into adjustable stimulation (AS), preset stimulation (PS), medical management (MM)), or an ancillary group who did not experience a response to an entry criterion of response to occipital nerve block, but did receive treatment as in the adjustable stimulation group. The average duration of symptoms was 22 (range 1–51) years. At three months, 39% of AS, 6% of PS, and 0% of MM subjects achieved a reduction of at least 50% in the number of headache days per month or a reduction in average overall pain intensity compared to baseline, which was statistically significant. However, there were no statistically significant differences in change in headache days or pain and duration between the AS and control groups [41]. 

Another RCT of eight patients implanted with bilateral ONS for migraine was conducted. These patients had had stable and significant pain relief with prior ONS. They were then randomly cycled between “effective”, “subthreshold”, and “no” stimulation. There were significant differences in improvement in VAS between the subthreshold and no stimulation groups, as well as between the effective and subthreshold groups, suggesting that paresthesias are not necessary for effective pain relief with ONS [42]. 

Another twenty patients with migraines who underwent successful (at least 50% reduction in pain between 3 and 5 days) ONS trials received permanent implantation and were randomized to either active or no stimulation for 12 weeks, followed by 40 weeks of active stimulation. After the twelve-week randomization period, 60% of patients in the active group reported at least a 30% reduction in pain in terms of VAS compared to only 20% of patients in the control group, though this result was not statistically significant. However, 30% of patients in the active group did report at least a 50% reduction in pain in terms of VAS compared to none in the control group, which was statistically significant. In addition, there was a significant difference between groups in mean daily VAS scores (control 67.85 ± 18.58 compared to active 29.98 ± 14.64). The change from baseline (control 59.94 ± 23.39 compared to active 50.94 ± 17.95) was also statistically significant. At the end of the 52 weeks/open phase of the experiment, 60% of patients reported at least a 30% reduction in VAS and 35% reported at least a 50% reduction, with an average daily VAS decrease by 16.9 mm from baseline for all patients (53.64 ± 19.55 to 36.74 ± 21.70) [43]. 

Here, 110 patients with migraine were randomized into four different treatment groups receiving different frequencies of transcutaneous ONS (tONS) (2 Hz, 100 Hz, 2/100 Hz—3 s of 2 Hz followed by 3 s of 100 Hz, sham) and one group receiving topiramate. All tONS with active stimulation and the topiramate group had a statistically significant difference when compared to sham with a 50% responder rate, but there was no significant difference between the tONS and topiramate groups. Compared to sham, only the 100 Hz tONS and topiramate groups displayed significant differences in terms of reduced headache days, though all five groups had significant differences between pre- and post-treatment measures. There was a significant difference noted between treatment and sham groups in regard to a decrease in headache intensity (measured via VAS), though again, there were significant decreases for all five groups pre- and post-treatment [44]. 

#### 2.1.7. Complicated Migraine, Occipital Neuralgia, Combined

A case of an individual with a dual diagnosis of complicated migraine, accompanied by phonophobia, temporary bilateral vision loss, slurred speech, ptosis, hemiplegia, and occipital neuralgia responding well to combined occipital nerve and bilateral subcutaneous temporal region stimulation was reported. After a seven-day trial of peripheral nerve stimulation, the patient underwent permanent implantation bi-temporally and over the inferior border of the temporalis muscle. At 24-month follow-up, the onset of headache saw a 50% reduction as well as the resolution of neurologic deficits associated with migraine at baseline [45].

#### 2.1.8. Primary Headache, Combined

In one prospective, uncontrolled feasibility study, nine patients with intractable CM or cCH were treated with ONS. Six of the nine patients had cCH, of which one also had migraines, and one also had HC. Three of the nine patients had CM alone. Eight out of nine patients completed the study. Overall, 62.5% of patients were deemed to have an excellent response, 25% were deemed to have fair responses, and 12.5% were deemed to have poor responses. There was a mean decrease of 28.5 in headache days and an average decrease in headache severity score of 0.88. Those with an excellent response had a greater than 90% reduction in disability (MIDAS), a greater than 50% reduction in headache days and/or a 30–50% reduction in average headache severity. Those with a fair response had a 25–50% reduction in headache days or severity [46].

#### 2.1.9. Craniofacial Pain

A review of prospectively collected data of thirty patients who had undergone neuromodulatory procedures for craniofacial pain was performed. In this study, 22/30 patients who had undergone the trial stimulation experienced a more than 50% reduction in pain intensity and went on to have permanent system implantation; three had infraorbital, four had supraorbital, thirteen had occipital, one had a combination of infraorbital and occipital, and one had a combination of supraorbital and occipital stimulation. The mean follow-up was 35 (range 1–77) months, with eighteen patients following up for more than a year and thirteen following for more than three. By the time of final follow-up, 22.7% had removed their implants, 9.1% due to improvement in pain intensity, 9.1% due to loss in efficacy, and 4.5% due to an infection. In total, 73% of the implanted 22 patients experienced a greater than 50% improvement in pain intensity, and 13.5% reported a less than 50% improvement. If one considers the total number of patients who had undergone the trial, 53% of those patients had a greater than 50% improvement with peripheral nerve stimulation [47].

### 2.2. Trigeminal Ganglion

The trigeminal ganglion is an area of particular interest in the pathophysiology of headaches and facial pain. It is the location of the primary sensory neuron of the trigeminal nerve and receives craniofacial sensory information from the trigeminal and occipital nerves. Due to its pathway along with the V1 branch of the trigeminal nerve in the nucleus of the solitary tract, sensory information of meninges may cause referred pain in the V1 area [48].

#### 2.2.1. Trigeminal Neuropathy

Several case reports have examined the effect of trigeminal (Gasserian) ganglion stimulation for the treatment of facial pain. In two case reports of trigeminal neuropathy, one presenting with facial pain and the other with facial itching, both patients had near total resolution of their symptoms within weeks to months after implantation of the nerve stimulator [49].

A study of 34 subjects found that 56% of patients had success with stimulation trials and progressed to permanent implantation of stimulators. Of that group, 53% had success of at least 50% pain relief, thus a total percent success rate of 29% among all patients. Of note, patients with central pain (e.g., secondary to stroke) had a higher success rate (71%) when compared to patients with pain of peripheral origin (e.g., previous surgery or facial trauma) (23%) [50].

In one retrospective case series of 59 patients with intractable facial pain, trial trigeminal stimulation produced positive results in about 70% of patients. Notably, facial trauma and history of oral surgery were predictors of failure [51]. 

Another retrospective analysis of 22 patients with trigeminal neuropathic pain found that 77.3% of patients had positive test stimulation, and of them, 88% reported initial pain reduction of at least 50%. In a further 6-month follow-up, 93.8% of patients had some level of pain relief, which decreased to 46.7% at 24 months [6].

In a larger study of 321 patients who had undergone trigeminal ganglion electrostimulation for trigeminal neuropathy, 52% of patients reported a pain reduction of at least 50% in a long-term follow-up, which ranged from anywhere from 5 to 25 years, with 82% reporting good or excellent analgesia. Of note, the results appeared to be particularly promising in posttraumatic patients as well as in patients with a history of maxillofacial or dental surgery. The results were not as promising in patients with postherpetic neuralgia [52].

#### 2.2.2. Trigeminal Neuralgia

In a study examining 267 patients with intractable pain from trigeminal neuralgia across eight clinical studies, 48% had at least 50% long-term pain relief. Patients with postherpetic neuralgia were reported to have a low success rate of 10% [53].

### 2.3. Supraorbital/Supratrochlear Nerve

The supraorbital nerve is a derivative of the frontal nerve, which is itself the first branch of the trigeminal nerve (V1/ophthalmic nerve), which is responsible for sensory innervation to the skin of the lateral forehead and upper eyelid, as well as the conjunctiva of the upper eyelid and mucosa of the frontal sinus [54]. The supratrochlear nerve is also a derivative of the frontal nerve. It is responsible for sensation to the cornea, conjunctiva, the skin of the upper eyelid, the bridge of the nose, and the skin of the forehead [55].

#### 2.3.1. Postherpetic Neuralgia

Two cases of individuals suffering from chronic, intractable postherpetic neuralgia (PHN) greater than four years responsive to treatment with peripheral nerve stimulation were described. In both patients, opioid medications, anticonvulsants, and antidepressants were ineffective in addressing the PHN, yet complete resolution was achieved upon region blockade of supraorbital and supratrochlear nerves with local anesthetics, which prompted the trial of peripheral nerve stimulation lead placement in the supraorbital nerve. Subsequently, permanent implantation was carried out, which resulted in a significant reduction in VAS score and the discontinuation of opioid medication [56].

A study examined the effect of combined occipital nerve and supraorbital nerve stimulation in fourteen patients experiencing chronic migraine refractory to conservative pharmacological modalities. The follow-up ranged from three to sixty months, and 71% of the patients included in the study saw a 50% or greater decrease in pain severity, with a mean VAS score reduction of 3.92 ± 2.4 [57]. 

Another study reviewed the data of ten patients with trigeminal neuropathic pain (TNP) after facial trauma or herpes zoster infection refractory to customary treatments who underwent peripheral nerve stimulation of the supraorbital or infraorbital branches of the trigeminal nerve. At the mean follow-up period of 26.6 ± 4.7 months post-transplantation of PNS, on average, pain relief, defined by at least a 50% reduction, was achieved in 70% of patients, decreased medication use was achieved in 70% of patients, and a satisfaction rate of 80% was achieved in terms of the patients [58].

#### 2.3.2. Cluster Headache

A case report of a 35-year-old male with intractable cluster headache demonstrated successful treatment with a unilateral percutaneous supraorbital nerve stimulation trial and permanent implantation [59]. 

A retrospective case series of sixteen patients who underwent a trial of supraorbital nerve stimulation, ten of whom had progressed to permanent implantation, found significant reductions in headache scores after five to seven days of stimulation, which persisted up to thirty weeks after permanent implantation. In addition, there was a statistically significant decrease in the morphine equivalent required for the relief of headaches. The mean duration of their headaches had been 10.6 (range 1.1–26.1) years [60].

A study of five patients, four with intractable cluster headaches and one with SUNCT syndrome, who were trialed and further implanted with supraorbital nerve stimulators, found an improvement in all patients in regard to functional status. VAS scores decreased from an initial mean of 8.9 to 1.6, and there was a reported improvement in the frequency and intensity of attacks [61]. 

Another seven patients with chronic and refractory migraine headaches had permanent combined occipital–supraorbital nerve stimulation systems implanted. The mean follow-up duration was 25.2 months. All patients reported improvements in functional status, pain intensity (VAS), and the frequency of headaches. The researchers examined both occipital stimulation and occipital–supraorbital dual stimulation. All subjects reported that the combined stimulation produced superior outcomes to occipital stimulation alone [62].

A multicenter RCT with 67 patients who were randomized into either transcutaneous supraorbital neurostimulation (tSONS) or sham for the prevention of migraines found a significant decrease in the mean number of migraine days between the intervention and control group, as well as a significant difference of 50% in terms of responder rate, monthly migraine attacks, monthly headache days, and monthly acute anti-migraine drug intake [63]. 

#### 2.3.3. Migraine

In a multicenter, double-blind, randomized, sham-controlled study, known as the “ACME” study, researchers recruited patients to receive either verum or sham external trigeminal nerve (supraorbital and supratrochlear) stimulation for one hour to treat acute migraine headache. In total, 106 patients were randomly assigned to either the verum or sham group, of which 99 completed the study. The primary outcome, mean change in pain score at 1 h compared to baseline, was significantly decreased in both groups but significantly more in the verum (−59%) than the sham (−30%) group. At later time points, 2 and 24 h, the difference in mean pain score reduction between groups remained significant. In addition, the proportion of pain-free subjects was significantly higher in the verum (29%) than sham (6%) group at 1 h but not 2 and 24 h follow-ups. This same pattern was observed for the proportion of subjects achieving at least 50% (63% vs. 31%) and 30% pain relief (79% vs. 39%) [64].

A study examined 24 patients with migraine without aura benefitting from transcutaneous supraorbital neurostimulation (tSNS). From the second month to the last month of the tSNS therapy, reduction in migraine attacks (at least a 50% reduction in 81% of patients) and migraine days per month (at least a 50% reduction in 75% of patients) as well as the reduction in the severity of headaches during the attacks were reduced (*p* < 0.05). The study suggests that tSNS may be considered a prophylactic treatment of migraine attacks in patients who are unwilling to comply with daily medications [65].

A randomized controlled study of 154 cases of individuals with episodic migraine headaches benefiting from combination therapy of flunarizine and transcutaneous supraorbital neurostimulation (tSNS) as a preventive treatment was conducted. The patients with episodic migraine were randomly assigned to a group of either flunarizine treatment of 5 mg per day, tSNS for 20 min daily, or combination therapy for three months. At the three-month follow-up, a reduction in days of experiencing migraine per month was achieved in all three groups; however, the result was more pronounced in the combination group, as evidenced by the 50% responder rate of 78.43% in the combination group in comparison to 46.15% in flunarizine monotherapy or 39.22% in tSNS [66].

#### 2.3.4. Hemiplegic Migraine

A study of four patients with hemiplegic migraine, a severe form of migraine marked by its motor aura, resistant to conventional therapies, responding to combined occipital and supraorbital neurostimulation, has been reported. Within a follow-up period ranging from 6 to 92 months post-transplantation of a peripheral nerve stimulator, on average, headache frequency decreased by 92% from 30 to 2.4 headache days per month, VAS reduced by 44% from 9.5 to 5.3, and the frequency of hemiplegic episodes diminished by 96% from 7.5 to 0.25 hemiplegic episodes per month [67].

#### 2.3.5. Supraorbital Neuralgia, Traumatic

The case of an individual with severe headaches stemming from supraorbital neuralgia benefitting from the peripheral neurostimulation of the supraorbital nerve was reported. The patient suffered a trauma to the right zygomatic arch-malar region, resulting in a frontal fracture. They initially received pharmacological treatment, which provided minimal relief to the pain. After temporary relief of pain was achieved via local anesthetic block of the supraorbital nerve, the patient underwent a 14-day trial with a peripheral nerve stimulator to the supraorbital nerve, which resulted in an improved VAS score. Encouraged by the results of the trial, a permanent implantation was administered that allowed the patient to gradually reduce the medication and ultimately discontinue its use [68].

### 2.4. Vagus Nerve

The vagus nerve (cranial nerve X) is a nerve involved in autonomic (parasympathetic), respiratory, cardiovascular, gastrointestinal, and pain systems [69]. Vagal afferents from the dura may also contribute to trigeminal pain due to their connection in the nucleus of the solitary tract; the vagus nerve may also contribute to trigeminal pain. 

#### 2.4.1. Non-Specific or Combination Primary Headache

Vagus nerve stimulation, both invasive and non-invasive, has been investigated for the treatment of primary headaches. In one case study, a 42-year-old male with a history of epilepsy and migraines underwent vagus nerve stimulation for the treatment of his seizures. It was found that stimulation was demonstrated to have some positive effect in reducing the frequency of his migraine attacks from 3 in one month to 3 in 13 months [70]. 

Another small study described case reports of six patients suffering from migraines (four) or cluster headaches (two) who received vagus nerve stimulation. The duration of symptoms ranged from 5 to 32 years. In total, 33% had an “excellent” response to VNS, and another 33% had a “good” response. Furthermore, 17% (both with migraines) never received any relief from stimulation, while another 17% had initial relief for three months but then had returned to baseline [71].

#### 2.4.2. SUNA

In an audit of an open-label prospective clinical study, researchers evaluated the preventive and abortive effects of nVNS in 41 patients with refractory primary chronic headaches (23 with CM (median age 44 years), 12 with cCH (median age 49.5 years), 4 with hemicrania continua (median age 47.5 years), 2 with short-lasting neuralgiform headache attacks with autonomic symptoms (SUNA) (median age 42 years). The patients were offered treatment with nVNS for three months. Neither of the patients with SUNA reported any noticeable improvement with nVNS use for three months for either abortive or preventive treatment [72].

#### 2.4.3. Hemicrania Continua

In the aforementioned study, 50% of patients with HC reported meaningful improvement with nVNS after three months. One had reported a 72.7% decrease in headache exacerbations at three months and a reduction of 1.8 points from 10/10 on NRS. However, by eleven-month follow-up, the exacerbations were still decreased (27.3% decrease) from baseline but higher than recorded at three months. The second patient had follow-ups at three, six, eight, and ten months, with exacerbations decreasing by 90%, 100%, 60%, and 80% at those time points, respectively. Headache intensity also decreased from 5.9/10 to 2.5, 4.2, 3.2, and 2.9, respectively [72].

#### 2.4.4. Cluster Headache

A specific form of vagal nerve stimulation, known as transcutaneous non-invasive vagal nerve stimulation (nVNS) (also known as gammaCore), has also been investigated as a treatment of cluster and migraine headaches.

Twelve patients with cCH in a previously mentioned study were treated with three months of nVNS. Regarding prophylactic efficacy, 8% noted at least a 30% reduction in weekly CH frequency at three months, as well as a reduction in oxygen use per week. Moreover, 17% had a slight improvement in weekly frequency, and 25% noted no changes to headache, while 50% noted worsening CH frequency after three months. In addition, none of the patients reported any abortive effect of nVNS on CH [72].

In an open-label cohort study, an audit was conducted of nineteen patients with cluster headache who were prescribed nVNS. In a 12-month period, nVNS was found to have utility in both acute and preventive therapy for cluster headaches. There was a reported overall mean improvement in the condition from baseline by 50% and an ability to abort nearly half of acute attacks within 15 min. In addition, there was a reported near 50% reduction in attack frequency in patients who used nVNS preventively [73].

A retrospective analysis was carried out for 30 patients (29 chronic and 1 episodic) with intractable CH who received nVNS for an evaluation period. The mean age was 47.9 (range 16–72) years, with a mean time since diagnosis of 7.2 (range 0–22) years. The mean duration of the evaluation period was 7.6 (0.9–27.5) months). In this study, 53% used nVNS exclusively as preventive therapy, 3% (the patients with episodic CH (eCH)) used it exclusively for acute treatment, and 43% used it for both. After beginning nVNS therapy, 10% of patients (all with cCH) were attack-free. Most patients were able to decrease or stop existing acute treatments while using nVNS therapy [74].

Several RCTs have been conducted to test the efficacy of nVNS in both cCH and eCH. In the “ACT1” randomized, double-blind, sham-controlled prospective study of 150 subjects (133 ITT) subjects randomized to either nVNS or sham treatment, there were no significant differences observed in a variety of endpoints, including response rate, sustained treatment response rate, and the percentage of patients who were responders, and of those who were pain-free, for at least 50% of the treated attacks. Interestingly, the results were significant when the subjects were analyzed separately for the eCH but not the cCH cohort [75]. 

In the “ACT2’’ study, a randomized, double-blind, sham-controlled prospective trial, 102 participants with either eCH or cCH were randomized to a sham or active nVNS group for two weeks and were then transitioned into an open-label period for two weeks. Between the treatment and sham groups, there was no statistically significant difference in the proportions of treated attacks that achieved pain-free status within fifteen minutes, but there was a significantly higher proportion of treated attacks that achieved pain-free status with nVNS than sham in the eCH group, which was not appreciated in the cCH group. In addition, the mean proportion of treated attacks per subject that achieved responder status (pain score of 0 or 1) within thirty minutes was greater with nVNS than sham in the total cohort but no subgroup individually. There was no inter-group difference appreciated for this endpoint. The mean decreases in pain intensity from attack onset to the 15- and 30-min time points were significantly greater in the nVNS arm of the eCH but not the cCH group or total cohort. The researchers noted the difficulty in accurately interpreting the data during the open-label period due to changes to participants’ prophylactic treatment regimens [76].

The “PREVA” trial was an open-label randomized study of 97 subjects (93 ITT) comparing nVNS with standard of care (SoC) medications compared to SoC alone for the treatment of cluster headaches. Patients were in a randomized phase for four weeks. There was a statistically significant reduction in the number of attacks per week compared to controls after randomization (−5.9 vs. −2.1), with a mean therapeutic gain of 3.9 fewer CH attacks per week. For those who participated in the four-week extension phase, there was an additional reduction in two CH attacks per week. The researchers also noted a significantly higher response rate, at least 50%, in the treatment than the control group, and there was evidence that those who remained in the study longer continued to respond to treatment. Furthermore, during the randomized phase, there was a 57% decrease in the frequency of abortive medication use in the treatment group that was not observed in the control group. There was also a significant improvement in quality of life for those undergoing treatment than those in the control [77]. 

#### 2.4.5. Migraine

Here, 23 patients with CM in a previously mentioned study were treated with three months of nVNS. Regarding prophylaxis, 8.7% of the patients reported at least a 30% reduction in the number of headache days, and 17.4% had less than a 30% reduction. Overall, 30.4% reported a worsening number of headache days, and 43.5% reported no change. No patients were able to abort migraines entirely with nVNS, though 8.7% were able to reduce pain severity. Furthermore, 47.8% had improvements in headache-related disability (as measured by HIT-6), which may be clinically significant, but were still at grade IV (considered severely disabled). Moreover, 4.3% experienced a reduction from grade IV to grade I on HIT-6, but all other patients had unchanged scores [72].

In one retrospective, systematic study, researchers asked 34 patients who had previously received VNS to fill out a questionnaire regarding headaches, specifically migraines, in the period of three months before VNS placement (period I), three months after (period II), and three months after that (period III). Of the thirty-four asked, five did not respond, and four were excluded due to cognitive deficits. In total, ten patients were identified as migraineurs (age 18–36 years). The mean delay between VNS placement and follow-up was 17 (range 4–36) months. In total, 80% reported a reduction in monthly frequency of at least 50%, and this occurred in the first three months after VNS placement and was maintained through the next three months. There was a statistically significant decrease in mean headaches per month between periods I and II but not between periods II and III. Overall, 50% of migraineurs reported becoming pain-free [78].

Another retrospective study was conducted to identify patients who had undergone VNS implantation for intractable epilepsy who also had concomitant chronic pain. Those identified were interviewed for pain intensity before VNS implantation and at the time of the interview. Of 62 patients who received VNS, 27 were contacted and of those 27, 4 had common migraines (ages 27–45 years). The duration of symptoms ranged from at least 8 to 15 years (not specifically stated in the study). At the time of the interview, 25% reported complete pain relief, 25% reported mild pain relief, and 50% reported a lot of pain relief. Furthermore, 75% also noted improvement in worst pain in terms of NRS, and 100% reported a decrease in migraine frequency [79].

Thirty subjects (27 ITT) with migraine (ten with and twenty without aura) participated in an open-label, single-arm, multiple-attack, pilot study. The median age was 39 years. The subjects were asked to nVNS specifically to treat up to four acute migraine attacks within six weeks. For the first attack, 21% of the subjects with baseline moderate or severe headaches reported being pain-free two hours after treatment, and 47% reported some pain relief. Furthermore, 63% of those with baseline mild pain were also pain-free two hours after treatment. For all attacks, subjects with baseline moderate or severe headache reported being pain-free two hours after treatment in 22% of attacks and some pain relief in 43% of attacks. For those with baseline mild pain, they reported being pain-free in 38% of attacks [80].

A three-month, single-center, open-label, prospective, observational cohort study was conducted to evaluate the impact of nVNS on the prevention and abortion of headaches in patients with intractable CM and EM. There were twenty participants, with a mean age of 53.1 (range 35–72) years, half with EM (nine without and one with aura) and half with CM (eight without aura and two with aura). Median pain VAS declined significantly in the total population from 8 (range 7.5–8) to 4 (range 3.5–5). The mean number of headache days per month decreased by a significant 39.5% in total. Migraine attacks per month decreased by 38.4%. When analyzed separately, both CM and EM groups had statistically significant improvements in median pain intensity, mean number of headache days per month, and migraine attacks per month. Moreover, 25% were pain-free within 2 h after starting adjunctive nVNS treatment at three months. For the total population, there were significant reductions in MIDAS score and grade, BDI, and Global Pittsburgh Sleep Quality Index (PSQI). These patterns persisted in individual group analysis, except for the fact that there was no significant decrease in MIDAS in the EM group [81].

In an open-label, single-arm, multicenter study, fifty patients (age 18–65 years) with high-frequency EM (fourteen patients) or CM (thirty-six patients) self-treated up to three consecutive mild or moderate migraine attacks during a two-week period with nVNS, though two did not treat any attacks. The mean disease duration was 29.7 years. Of 48 patients, 56.3% reported pain relief, defined as at least a 50% reduction in VAS at one hour. Of this group, 62.9% (35.4% of the total) reported being pain-free. Moreover, 64.6% reported pain relief at two hours. Of this latter group, 61.3% (39.6% of total) were pain-free. Of all 131 recorded migraine attacks, there was pain relief in 38.2% and 51.1% of attacks at one and two hours, respectively. In 17.6% and 22.9% of attacks at one and two hours, respectively, there was full pain-free status. The researchers note that a greater proportion of EM than CM patients had pain relief and pain-free status at both time points [82].

There have been several RCTs examining nVNS on migraine headaches. One, known as the “PRESTO” study, included a total of 248 participants (243 ITT) with episodic migraine with and without aura to receive either nVNS or sham within twenty minutes of pain onset. Although pain-free responder rates were absolutely higher for nVNS as compared to sham at 30 and 60 min, they did not reach significance. Interestingly, at 120 min, they were statistically significant. nVNS was also superior to sham in abortion of the first treated attack at 30 and 60 min, but not at 120 min, though a repeated-measures test found nVNS to be superior to sham for the pain-free outcome across all time measurements [83]. 

In the “PREMIUM I” trial, a prospective, multicenter, randomized, double-blind, parallel-group, sham-controlled RCT with 477 patients (332 ITT), there was no statistically significant difference in the number of migraine days per month, the percentage of patients with at least a 50% reduction in the number of migraine days, mean reductions in the number of headache days, nor reductions in the number of acute medication days between the nVNS and sham groups after a 12-week double-blind period. On further analysis of modified ITT (population who were adherent at least 67% of the time), there were significant therapeutic gains in terms of migraine days, headache days, and acute medication days between groups. The researchers noted that the sham may have had some level of active vagus nerve stimulation, which may have contributed to the lack of statistically significant findings in the trial. Overall, 269 (187 ITT) patients entered a 24-week open-label period, and therapeutic benefits in terms of migraine and headache days persisted [84].

Another RCT, known as the “PREMIUM II” trial, also examined nVNS in relation to migraines with and without aura. With 336 participants assigned to both nVNS and sham during a 12-week double-blind period, this study also found that the rate of participants with at least a 50% reduction in the number of migraine days was statistically significantly greater in the nVNS than in the sham group, though the decrease in the number of migraine days from baseline did not reach statistical significance between the groups. However, nVNS was significantly better at decreasing headache impact (HIT-6) and disability (MIDAS), and a greater proportion of patients in the treatment arm were satisfied with treatment. The effects were more pronounced for those with aura than those without [85]. 

Another prospective, multicenter, double-blind, sham-controlled pilot study of nVNS in CM prophylaxis, known as “EVENT,” with 73 participants (59 ITT), was conducted. In this study, 51 participants from the ITT and 49 from the ITT completed a 2-month randomized phase; 48 participants from the ITT and 47 from the PP continued into an open-label phase for six months with follow-up every two months. Overall, 27 participants completed the study. Between groups, there was no statistically significant difference in the number of headache days. However, after the open-label phase, participants who had initially been assigned to the nVNS group had a statistically significant decrease in mean headache days after eight months. The researchers noted that with longer follow-up, the patients had improved results, with more participants in the treatment arm achieving at least a 50% response compared to none in the control [86].

Another monocentric, prospective, double-blind RCT was conducted with 46 participants suffering from CM. Here, 46 participants (ITT) were randomized into a 1 Hz or 25 Hz group, but 39 were included in the per-protocol (PP) analysis. PP analysis demonstrated a 36.4% reduction from baseline in headache days per 28 days in the 1 Hz group and a 17.4% reduction in the 25 Hz group. The reduction from baseline in both groups was statistically significant as was the difference between the groups. The ITT analysis demonstrated a significant decrease in both groups but no statistically significant inter-group differences. Furthermore, 29.4% of participants in the 1 Hz (PP) and 13.6% in the 25 Hz group (PP) reported a greater than 50% improvement in headache days, which was not statistically significant from baseline, nor was it significant between groups. However, the number of days with intake of acute headache medication as well as MIDAS and HIT-6 scores were significantly reduced in both treatment groups, without significant differences between the groups [87].

### 2.5. Cervical Spinal Cord

Cervical spinal cord stimulation is often used in the treatment of upper extremity radicular pain, complex regional pain syndrome, neuropathic pain syndromes, and post-laminectomy pain syndrome; however, only several clinically meaningful studies and cases have been reported that suggest the role of cervical spinal cord neuromodulation in the treatment of headache and facial pain, let alone the proposed mechanism that would explain the efficacy [88].

#### 2.5.1. Migraine

In one prospective, long-term, single-center, open-label study, 20 patients with refractory CM underwent implantation of a 10 kHz spinal cord stimulator, with the distal tip of the leads positioned at the C2 vertebral level. The patients were followed up at 52 weeks post-implantation. In comparison to the initial evaluation, 60% of patients reported at least a 30% reduction in mean monthly migraine days (MMDs), and 50% of patients saw at least a 50% reduction in mean MMD. In addition, the nature of the headache transitioned from chronic to episodic in 50% of the patients at 52 weeks [89].

#### 2.5.2. Cluster Headache

In another study, seven patients with cCH were subcutaneously implanted with high cervical epidural electrodes after a median test phase of 10 (range 4–19) days, followed up in 23 months on average (range 3–78 months). Compared to the baseline mean frequency of attacks of 5.0 attacks/day (range 1.7–10.0 attacks/day), post-implantation attack frequency decreased to 1.4 attacks/day (0–3.5 attacks/day). In addition, the mean intensity score (VAS) of 7.4/10 (range 4.3–10.0) was reduced to 4.5/10 (range 0–7.6) upon follow-up [90].

#### 2.5.3. Post-Traumatic Headache

A case was described in which an individual suffered from a post-traumatic cervicogenic headache with disc herniation. His symptoms were inadequately controlled by medication management alone. After successful blocks of the greater occipital nerve and C3 roots, he underwent a temporary implantation of a spinal cord stimulation electrode at the cephalad end of the C3–C4 vertebral level, which provided an 80% improvement in terms of neck pain and the resolution of supraorbital pain [91].

#### 2.5.4. Postherpetic Neuralgia

Another case was reported on the use of short-term high cervical spinal cord stimulation at the C1–C2 level in the treatment of refractory trigeminal postherpetic neuralgia affecting the V2 and V3 distributions. Electrical stimulation was given in tonic mode with a pulse width of 450 microseconds, a frequency of 40 Hz, and a constant current amplitude of 3 mA, which resulted in paresthesia in the V3 distribution but not in V2. Compared to the baseline NRS score of 7–9, the post-implantation score decreased to 1–2, with a marked improvement in quality of life [92].

### 2.6. Infraorbital Nerve

The infraorbital nerve is a branch of the second division of the trigeminal nerve, also known as the maxillary nerve. Blockade with local anesthetic has been shown to relieve headache in both acute and chronic settings, as suggested by a study of 26 patients suffering from migraine seeing improvements in mean MIDAS and VAS scores after supraorbital and infraorbital nerves block [93].

#### Cluster Headache

A case of a patient suffering from chronic, refractory, recurrent cluster headache was reported who responded well to the combined stimulation of occipital, supraorbital, and infraorbital nerves with a peripheral nerve stimulator. At 36-month follow-up post-implantation of the stimulator, the frequency of the headaches dramatically reduced from 3–4 episodes per day to 3–4 per month. Though the case describes the simultaneous stimulation of three different peripheral nerves and, therefore, cannot be used to assess the efficacy of an isolated nerve, the case is significant in its improvement upon long-term follow-up [94].

### 2.7. Great Auricular Nerve

The great auricular nerve (GAN) is a pure sensory nerve that is one of the superficial branches of the cervical plexus, arising from the second and third cervical nerves coursing around the posterior border of the sternocleidomastoid muscle. The GAN communicates with the lesser occipital nerve, the auricular branch of the vagus nerve, and the posterior auricular branch of the facial nerve [95].

#### 2.7.1. Migraine

A case of a migraine headache intractable to medical management resulting in sustained pain reduction after the implantation of a great auricular nerve stimulator was described. After failure to alleviate the headache with an exhaustive list of oral medications, GAN block was trialed over three different attempts, all of which the patient saw immediate pain relief on each occasion. Subsequently, the patient underwent permanent GAN stimulator implantation, and at six-month follow-up, the patient reported significant alleviation of pain [95].

#### 2.7.2. Post-Traumatic Headache

A case of a post-traumatic headache intractable to years of medical management responding to GAN neuromodulation was reported. After several attempts to abort headache attacks with blocks of GAN, a trial of subcutaneous electrode placement over the C2–C3 branches of GAN was carried out. Then, the patient underwent permanent implantation of a stimulator for six months. The patient saw a greater than 90% reduction in headache frequency upon follow-up [96]. 

### 2.8. Auriculotemporal Nerve

The auriculotemporal nerve (ATN) is the last branch of the third division of the trigeminal nerve, also known as the mandibular nerve, often implicated in ipsilateral preauricular area pain, the attacks of which are often paroxysmal and ranging from moderate to severe as well as in migraine headaches [97,98].

#### 2.8.1. Migraine

A case of severe, refractory CM triggered by loud noises since childhood alleviated through the application of neuromodulation was reported. The patient underwent a three-week trial of bilateral percutaneous placement of peripheral nerve stimulators targeting the auriculotemporal nerves, which precipitated at least a 50% reduction in the intensity of the headaches. Subsequently, a permanent implantation of a peripheral nerve stimulator was carried out in the bilateral auriculotemporal nerves, which was followed up at 16 months post-implantation. Upon follow-up, MIDAS had gone down drastically from grade IV (severe disability) to grade II (mild disability) and average pain intensity decreased from 8–9/10 to 5/10 [99].

#### 2.8.2. Chronic Headache

A study examined the cases of twenty-four patients who suffered from chronic headache and underwent peripheral nerve stimulator implantation after a successful trial phase. After implantation, all patients demonstrated a statistically significant improvement in head pain intensity, duration, and frequency at three-month follow-up period, as evidenced by a significantly decreased mean total pain index (TPI) from 516 ± 131 pre-implantation to 74.8 ± 61.6 post-implantation [100].

### 2.9. Sphenopalatine Ganglion

The sphenopalatine ganglion (SPG) is the largest aggregate of sensory, sympathetic, and parasympathetic nerves inside the calvarium apart from the brain itself and has been implicated in headache via the induction of trigeminal autonomic cephalalgia and facial pain via connection with facial, lesser occipital, and cutaneous cervical nerves. Therefore, numerous studies and cases have been reported that, in aggregate, could delineate the role of sphenopalatine ganglion neuromodulation in addressing headache and facial pain [101].

#### 2.9.1. Migraine

A study explored the role of SPG stimulation in eleven patients who suffered from intractable migraine headaches, which were unresponsive to the medical treatment, as the sphenopalatine ganglion has been suggested to be a switching nucleus for autonomic fibers. Upon the placement of the electrode, stimulation was given for an average of three minutes (1–4 min) with a mean amplitude of 1.2 V (0.9–1.8 V), a mean pulse rate of 57 Hz (50–120 Hz), and a mean pulse width of 394 ms (300–700 ms). Of the eleven patients, two patients experienced a complete resolution of migraine headaches within three minutes of stimulation, three patients saw a reduction in pain, five had no response, and one was not stimulated [102].

#### 2.9.2. Cluster Headache

The role of sphenopalatine ganglion stimulation in treating patients with cluster headache was described in an open-label follow-up study. A total of 33 patients were enrolled in the study; 45% of the participants were acute responders, defined as acute effectiveness in ≥50% of attacks or a ≥50% reduction in attack frequency when compared to the baseline. In these acute responders, of the 5956 attacks (180.5 ± 344.8 attacks), 4340 attacks were treated, 78% of which were due to SPG stimulation alone [103]. 

Another double-blind, randomized controlled trial also examined the efficacy of SPG stimulation on chronic cluster headache. The study included 93 patients aged 22 years or older who experienced four or more cluster headache attacks per week and were non-responsive to conventional preventive therapeutics, 45 of whom were assigned to the SPG stimulation group and 48 to the control group. Pain relief from headache attacks was achieved in 62.46% of the SPG stimulation group, while it was achieved in 38.87% of the control group [104].

#### 2.9.3. Facial Pain

A case was described of a 30-year-old suffering from right facial pain for more than nine years. Prior to the presentation, the patient was trialed on multimodal treatment, including antiepileptic medication, which provided minimal relief. Three attempts of SPG blocks using different local anesthetic medications were carried out, all of which resulted in significant temporary relief. The response to SPG blocks prompted the decision to proceed with the electrical nerve stimulation trial and subsequent permanent implantation of the electrode in the pterygopalatine fossa. The patient saw a dramatic response, with the complete discontinuation of opioid-based medication [105].

## 3. Complications

Taking the cited studies into consideration, the most common complications are hardware-related, of which the majority are lead migration or battery depletion, but also include the malfunction of the electrodes. Though not as common as hardware complications, biologic events, commonly pain at the site of the implantable pulse generator (IPG) site or infections can also occur [16,18,19,22,23,26,27,31,32,33,34,36,37,39,40,41,43,87].

A retrospective analysis of a prospective, multicenter, double-blind controlled study on ONS for the management of chronic migraine aimed to analyze adverse event incidence rate’s relationship with device characteristics, surgical techniques, and IPG placement was carried out. The researchers noted that IPG pocket locations closer to the lead and more experienced implanters were associated with lower complication rates [106].

## 4. Discussion

The results of neurostimulation in the context of headaches and facial pain are promising, especially in patients with pain refractory to other treatment modalities. In addition, given the significant suffering that patients with these conditions may deal with, neuromodulation is a way to provide significant relief to many patients who suffer from these conditions. The use of neurostimulation may also help reduce or eliminate the need for preventive medications in patients as well as improve disability, quality of life, mood, and sleep. However, many of the studies published as of the writing of this review are uncontrolled studies or case reports, which raise concerns for placebo effect and bias. This is in part due to practical considerations, such as the difficulty in blinding when many treatments elicit paresthesia and the relative rarity of some headache disorders, which was recognized by several authors. The crossover study design does mitigate this somewhat and, for this reason, many trials above have exploited it in their studies.

There is also a question of the long-term efficacy of neuromodulation. Many studies do show promise in some patients up to a few months out from implantation, but some do appear to experience the recurrence of headaches after some time. However, this may be secondary to technical failures, as some studies found that effective stimulation was not being delivered at the time of follow-up. Interestingly, there appear to be several patients reported in the cited studies who had had the exact opposite and found that they were able to explant permanent stimulators due to significant pain relief. A summary of cited studied are provided in Table 1.

In 2013, the European Headache Federation released a position on the use of neuromodulation for chronic headaches. Based on the available evidence at the time, it had recommended neurostimulation only in cases of chronic, medically intractable headaches. In the treatment of cCH, it recommended SPG or ONS prior to other forms of neurostimulation, such as DBS and in the management of CM, it stated that ONS may be acceptable. It did acknowledge that RCTs were scarce and that further trials may change the position on their use [107]. In 2021, the American Headache Society also stated that patients with intractable migraines or those with poor tolerability or contraindications with medications should be considered for a trial with FDA-approved neuromodulatory devices. For preventive treatment, they stated that all patients should be considered for a trial as an adjunct to the existing treatment plans [108].

A meta-analysis conducted in 2022 analyzed 45 studies studying preemptive treatments for refractory cCH, most focusing on neuromodulation. Consistent with the European Headache Federation’s recommendation years prior, the authors note that ONS appeared to be promising for the treatment of cCH. They also noted that DBS showed promise but had more heterogeneous results and a higher risk of complications [109]. Another meta-analysis analyzing 38 articles focused on acute or preventive treatment of migraines also noted the efficacy of ONS in terms of preventive migraine treatment [110].

## 5. Conclusions

The neurostimulation of various targets is a promising treatment for patients with medically intractable headaches. There is a need for further study, including high-quality RCTs. As of now, the literature is asymmetric, with some nerve targets (e.g., occipital and vagus nerves) undergoing many more trials than others. Neuromodulation should be considered for patients, especially those with medically intractable diseases or with significant impacts on functional ability and/or quality of life.

## Figures and Tables

**Table 1 biomedicines-11-03328-t001:** Summary of cited articles categorized by targeted nerves.

Occipital Nerve						
Author	Study Type	Condition	Targeted Nerve(s)	Number of Participants	Type of Intervention	Outcome
Weiner et al. [14]	Prospective	Occipital Neuralgia	Occipital	13	PNS	67% of patients with >75% pain relief; 33% of patients with >50% pain relief
Rajat et al. [15]	Case	Occipital Neuralgia	Occipital	1	PNS	0/10 pain scale at five- and twelve-month follow-ups
Salmasi et al. [16]	Case	Occipital Neuralgia	Occipital	3	PNS	55% (range 25–90%) pain relief at nine (range 6–16 months) months
Oh et al. [17]	Technical Report	Occipital Neuralgia	Occipital	10	PNS (C1-3 paddle-style electrode)	At one month, 80% patients with >90% pain reduction; 20% patients with 75–90% pain reduction; at six months, 78% patients with >90% pain reduction, 11% patients with 50–75% pain reduction
Kapural et al. [18]	Case-series	Occipital Neuralgia	Occipital	6	PNS	71% decrease in VAS and 72% decrease in pain disability index scores
Slavin et al. [19]	Retrospective	Occipital Neuralgia	Occipital	14	PNS	At mean follow-up of 22 months (range five-32 months), 50% patients pain-free, 30% with significant pain relief
Raoul et al. [20]	Retrospective	Occipital Neuralgia	Occipital	60	PNS	72.2% decrease in VAS and 72.7% decrease in number of headache days per month after one year
Miller et al. [21]	Prospective	Short-lasting Unilateral Neuralgiform Headache (SUNCT syndrome)	Occipital (bilateral)	31	PNS	At mean follow-up of 44.9 months (range 13–89 monts), 69% decrease in mean daily attack frequency; 38.7% of patient pain-free at the final follow-up
Pascual et al. [22]	Crossover	Hemicrania Continua	Suboccipital	6	PNS	At median follow-up of 13.5 months (range 6–21 months), 67% patients with 80–95% improvement, 17% patients with 30% improvement, and 17% patients with worsening pain by 20%
Balossier et al. [23]	Retrospective	Trigeminal Neuralgia	Occipital	7	PNS	Mean follow-up of 59 months (range 2–106 months); mean pain relief of 86.7% (range 70–100%); at last follow-up, mean pain relief of 58% (range 0–100%)
Mueller et al. [24]	Prospective	Cluster Headache	Occipital	10	PNS	90% patients with improvement; mean overall improvement of 44% (range 20–90%) of attacks; 30% patients with up to 19 days of pain-free period
Magis et al. [25]	Prospective	Cluster Headache	Suboccipital	8	PNS	Mean follow-up of 15.1 (range 3-22 months) months; 25% patients pain-free at 16 and 22 months, 38% patients with 90% attack frequency reduction, 25% patients with 40% improvement
Magis et al. [26]	Prospective	Cluster Headache	Suboccipital	15	PNS	Mean follow-up of 36.82 months; 80% patients with 90% improvement in pain, 60% patients pain-free, 13% patients with no or mild improvement
Leone et al. [27]	Monocentric, Open-Label	Cluster Headache	Occipital	35	PNS	Median follow-up of 6.1 (range 1.6–10.7 years) years; decrease in mean number of daily attacks from 5.7 to 2.4; 66.7% patients with >50% reduction in number of headache per day
Burns et al. [28]	Prospective	Cluster Headache	Occipital (bilateral)	14	PNS	Median follow-up of 17.5 (range four-35 months) months; 71% patients with improvement in frequency, severity, or duration; 30% patients with 90% improvement, 30% patients with >40% improvement, 40% patients with 20–30% improvement
Burns et al. [29]	Prospective	Cluster Headache	Occipital (bilateral)	8	PNS	Median follow-up of 20 (range eight-27 months) months; 75% patients with improvements; 25% patients with >90% improvement in attacks, 38% patients with >40% improvement, 13% patients with >25% improvement
Fontaine et al. [30]	Cohort	Cluster Headache	Occipital (bilateral)	13	PNS	Mean follow-up of 14.6 (range three-34 months) months; 77% patients with improvements within a few days of implantation; eight patients with 68% decrease in mean attack frequency and 49% decrease in mean attack intensity on NRS at one-year follow-up
Garcia-Ortega et al. [31]	Retrospective	Cluster Headache	Occipital	17	PNS	60% patients with complete remission, 100% patients with >80% decrease in attack frequency with a mean reduction of 42% in residual headache intensity
Miller et al. [32]	Prospective	Cluster Headache	Occipital (bilateral)	51	PNS	Mean follow-up of 39.17 months; 52.9% patients with >50% decrease in daily attack frequency, 47.1% patients with six months of continuous pain relief; of 27 patients using preventive medications at baseline, four ceased all preventive medication use and 17 reduced use
Tepper et al. [33]	Prospective	Cluster Headache	Occipital (bilateral)	67	PNS	At three-month follow-up, 75% patients with >30% reduction in attack frequency; at 12 months follow-up, 64% patients with >30% reduction in attack frequency and 59% patients with >50% reduction
Wilbrink et al. [34]	Randomized Controlled Trial (multicenter, double-blind, phase 3)	Cluster Headache	Occipital	131	PNS	Ten-day run = in phase, 24-week treatment period; Decrease in median weekly mean attack frequency (MAF) from 7.38 to 5.21 after ONS onset; between weeks 21–24 and 45–48, about 50% of patients with >50% pain relief
Popeney et al. [35]	Case Series	Migraine	Occipital	25	PNS	Mean follow-up of 18.3 (range nine-36 months) months; Mean headache frequency reduction from 75.6 days to 37.5 days; mean severity on a NRS scale of 0–10 decreased from 9.32 to 5.72
Oh et al. [17]	Prospective	Migraine	Occipital	10	PNS (C1-3 paddle electrode)	At one-month follow-up, 90% patients with >90% pain reduction, 10% patients with 75–90% reduction in pain; at six-month follow-up, 78% patients with >90% pain reduction, 22% patients with 75–90% pain reduction
Rodrigo et al. [36]	Prospective	Migraine	Occipital	41	PNS	Mean follow-up of 9.4 years; VAS decrease by 4.9, the number of migraine days per month reduction by 59.2%; 13.9% patients with permanent implantation pain-free at last visits
Ashkan et al. [37]	Prospective	Migraine	Occipital	112	PNS	At three-month follow-up, mean reported pain relief of 25.1% and 21.7% patients with >50% pain relief; at nine- and 12-month follow-ups, reported pain relief of 30.8%, and 28.5% patients with >50% relief
Schwedt et al. [38]	Retrospective	Migraine	Occipital (eight bilateral; seven unilateral)	15	PNS	Mean follow-up of 17.8 (range 6–36 months) months; severity reduction from 6.75 to 4.5 on a 0–10 scale; MIDAS from 170 to 116; HIT-6 from 73 to 61; BDI-II from 19 to 11.5
Matharu et al. [39]	Case Series	Migraine	Suboccipital (bilateral)	8	PNS	Mean follow-up of 1.5 years (range seven months to three years); 50% patients pain-free with only rare breakthrough headaches; 25% with very good responses with suppression of headaches
Serra et al. [40]	Prospective, randomized cross-over	Migraine	Occipital	34	PNS	Trial period of 15–30 days; Average follow-up of 45 days (range 12 to 122 days); 9% patients with >50% reduction in the number of attacks, 22% patients with >50% reduction in the severity of attacks, 66% patients with >50% reduction in both number and severity
Silberstein et al. [41]	Randomized Controlled Trial (double-blind)	Migraine	Occipital	268	PNS	157 subjects with permanent implantation; 59.5% patients > 30% reduction in headache days and/or pain intensity, 47.8% patients > 50% in headache days
Saper et al. [42]	Randomized Controlled Trial	Migraine	Occipital	75	PNS	At three months follow-up, 39% of adjustable stimulation (AS) group, 6% of preset stimulation (PS) group, and 0% of medical management (MM) group with reduction of >50% in number of headache days per month
Slotty et al. [43]	Randomized Controlled Trial	Migraine	Occipital (bilateral)	8	PNS	No significant differences in improvement of VAS between “effective”, “subthreshold”, and “no” stimulation from the prior ONS
Mekhail et al. [44]	Randomized Controlled Trial	Migraine	Occipital	20	PNS	Randomized to either active or no stimulation for 12 weeks, followed by 40 weeks of active simulation; after twelve week randomization period, 60% active group with >30% reduction in VAS, 20% control group with >30% reduction in VAS; at the end of 52 weeks/open phase, 60% patients with >30% reduction in VAS, 25% patients with >50% reduction in VAS
Liu et al. [45]	Randomized Controlled Trial	Migraine	Occipital	110	PNS	Randomized into transcutaneous ONS (2 Hz, 100 Hz, 2/100 Hz, sham) and treatment with topiramate; only the 100 Hz tONS and topiramate groups with significant differences in reduction of headache days compared to sham
Deshpande et al. [46]	Case	Complicated Migraine, Occipital Neuralgia, combined (phophobia, temporary bilateral vision loss, slurred speech, ptosis, hemiplegia)	Occipital, bilateral subcutaneous temporal region	1	PNS	At 24-month follow-up, 50% reduction in the onset of headache and resolution of neurologic deficits
Trentman et al. [47]	Prospective	Primary Headache, combined	Occipital	9	PNS	62.5% patients with excellent, 25% with fair, and 12.5% with poor responses; mean decrease of 28.5 in headache days, average decrease in headache severity score of 0.88
Slavin et al. [48]	Prospective	Craniofacial Pain	Infraorbital	30	PNS	22/30 patients with 50% reduction in pain intensity, of whom three had infraorbital, four had supraorbital, 13 had occipital, and one had a combination of infraorbital and occipital, and one had a combination of supraorbital and occipital stimulations; mean follow-up of 35 months (range 1–77 months); 73% of 22 patients with implantation with >50% improvement in pain intensity and 13.5% with <50% improvement.
**Trigeminal Ganglion**						
**Author**	**Study Type**	**Condition**	**Targeted Nerve(s)**	**Number of Participants**	**Type of** **Intervention**	**Outcome**
Logghe et al. [50]	Case	Trigeminal Neuropathy	Trigeminal Ganglion	2	PNS	Total resolution of symptoms within weeks to months after implantation in both patients
Taub et al. [51]	Prospective	Trigeminal Neuropathy	Trigeminal Ganglion	34	PNS	Of the group with permanent implantation, 53% patients with >50% pain relief
Texakalidis et al. [52]	Retrospective Case Series	Trigeminal Neuropathy	Trigeminal Ganglion	59	PNS	70% patients with positive results with a trial of trigeminal stimulation; history of oral surgery, facial trauma predictive of failure
Kustermans et al. [6]	Retrospective	Trigeminal Neuropathy	Trigeminal Ganglion	22	PNS	At the initial follow-up, 88% patients with >50% reduction of pain; at six-month follow-up, 93.8% patients with some level of pain relief; at 24 months, 46.7% patients with some level of pain relief
Mehrkens et al. [53]	Prospective	Trigeminal Neuropathy	Trigeminal Ganglion	321	PNS	At a long-term follow-up (range five to 25 years), 52% patients with >50% pain reduction, 82% patients with good or excellent analgesia
Holsheimer et al. [54]	Prospective	Trigeminal Neuralgia	Trigeminal Ganglion	267	PNS	48% patients with >50% long-term pain relief; patients with postherpetic neuralgia with low success rate of 10%
**Supraorbital/Supratrochlear Nerve**						
**Author [Citation]**	**Study Type**	**Condition**	**Targeted Nerve(s)**	**Number of Participants**	**Type of** **Intervention**	**Outcome**
Dunteman et al. [57]	Case	Postherpetic Neuralgia	Supraorbital	2	PNS	Significant reduction in VAS score and discontinuation of opioid medication
Hann et al. [58]	Prospective	Postherpetic Neuralgia	Supraorbital, Occipital, combined	14	PNS	Follow-up ranging from three to 60 months; 71% patients > 50% reduction in pain severity with mean VAS score reduction of 3.92
Johnson et al. [59]	Prospective	Postherpetic Neuralgia	Supraorbital, Infraorbital	10	PNS	At the mean follow-up of 26.6 months post-transplantation, 70% patients with >50% reduction in pain and decreased medication use
Narouze et al. [60]	Case	Cluster	Supraorbital	1	PNS	Successful relief and resolution of symptoms after implantation
Amin et al. [61]	Retrospective	Cluster	Supraorbital	16	PNS	Ten patients with permanent implantation; significant reduction in headache scores after five to seven days of stimulation, persisting up to thirty weeks; significant decrease in morphine equivalent required
Vaisman et al. [62]	Prospective	Cluster	Supraorbital	5	PNS	VAS scores decreased from 8.9 to 1.6; reported improvement in frequency and intensity of attacks
Reed et al. [63]	Prospective	Cluster	Supraorbital, Occipital, combined	7	PNS	Mean follow-up of 25.2 months; all patients with improvements in functional status, VAS, and frequency of headaches
Schoenen et al. [64]	Randomized Controlled Trial (multicenter)	Cluster	Supraorbital	67	PNS	Significant decrease in mean number of migraine days, difference in 50% responder rate, monthly migraine attacks, monthly headache days, monthly acute anti-migraine drug intake
Chou et al. [65]	Randomized Controlled Trial (multicenter, double-blind)	Migraine	Supraorbital, Supratrochler, combined	106	PNS	At one, two, and 24 h, mean change in pain score significantly reduced in the verum group compared to the sham; 29% of verum group pain-free, while 6% of sham group pain-free
Russo et al. [66]	Prospective	Migraine	Supraorbital	24	PNS	81% patients with >50% reduction in migraine attacks, 75% patients with >50% reduction in migraine days per month
Jiang et al. [67]	Randomized Controlled Trial	Migraine	Supraorbital	154	PNS	At three-month follow-up, more pronounced reduction in days of migraine per month in the combination group of flunarizine with neurostimulation; 50% responder rate of 78.43% in combination group, 46.15% in flunarizine monotherapy, and 39.22% in supraorbital stimulation monotherapy
Reed et al. [68]	Prospective	Hemiplegic Migraine	Supraorbital, Occipital, combined	4	PNS	Follow-up period running 6 to 92 months post-transplantation; mean headache frequency by 92% from 30 to 2.4 headache days per month; VAS reduction by 44% from 9.5 to 5.3; frequency of hemiplegic episode reduction by 96% from 7.5 to 0.25
Asensio-Samper et al. [69]	Case	Supraorbital Neuralgia, Traumatic	Supraorbital	1	PNS	14-day trial resulting in improvement of VAS score; permanent implantation to discontinue medications
**Vagus Nerve**						
**Author**	**Study Type**	**Condition**	**Targeted Nerve(s)**	**Number of Participants**	**Type of** **Intervention**	**Outcome**
Sadler et al. [71]	Case	Non-Specific/ Combination Primary Headache; history of epilepsy	Vagus	1	PNS	Reduction in migraine attack frequency from three in one month to three in 13 months
Mauskop et al. [72]	Case	Migraine (4), Cluster (6) Headaches	Vagus	6	PNS	33% patients with “excellent” response, 33% with “good” response to vagus nerve stimulation
Trimboli et al. [73]	Prospective	Short-Lasting Neuralgiform Headache Attacks with Autonomic Symptoms (SUNA)	Vagus	2	PNS	At three-month follow-up, the patients with SUNA reported noticeable improvement
Trimboli et al. [73]	Prospective	Hemicrania Continua	Vagus	4	PNS	At three-month follow-up, 50% of the patients with HC reported meaningful improvement; one patient with 72.7% decrease in headache exacerbations at three months
Trimboli et al. [73]	Prospective	Cluster Headache	Vagus	12	PNS	At three-month follow-up, 8% of patients with >30% reduction in weekly CH frequency and reduction in oxygen use per week, 17% patients with slight improvement in weekly frequency, 25% patients with no changes to headache, and 50% with worsening of CH frequency
Nesbitt et al. [74]	Cohort, Open-Label	Cluster Headache	Vagus	19	PNS	Mean improvement in condition from baseline by 50%; a near 50% reduction in attack frequency in patients using nVNS preventively
Marin et al. [75]	Retrospective	Cluster Headache	Vagus	30	PNS	53% patients used as preventive therapy, 3% patients used exclusively for acute, and 43% patients for both; mean follow-up of 7.6 (range 0.0 to 27.5 months) months; 10% patients free of attack
Silberstein et al. [76]	Randomized Controlled Trial (double-blind)	Cluster Headache	Vagus	150	PNS	No significant differences in response rate, sustained treatment response rate, the percentage of patients who were responders, and of those who were pain relief for at least 50% of treated attacks
Goadsby et al. [77]	Randomized Controlled Trial (double-blind)	Cluster Headache	Vagus	102	PNS	Randomized to a sham or active nVNS group for two weeks, then an open-label period for two weeks; significantly higher proportion of treated attacks that achieved pain-free status in nVNS than that of sham in the episodic CH group
Gaul et al. [78]	Randomized Controlled Trial (open-label)	Cluster Headache	Vagus	97	PNS	At four-week follow-up, statistically significant reduction in the number of attacks per week in nVNS group compared to the control (−5.9 vs. −2.1); during the randomized phase, 57% decrease in the frequency of abortive medication use in the treatment group
Trimboli et al. [73]	Prospective	Migraine	Vagus	23	PNS	8.7% patients with >30% reduction of headache days, 17.4% with <30% reduction, 30.4% patients with worsening number of headache days, and 43.5% patients with no change
Lenaerts et al. [79]	Retrospective	Migraine	Vagus	34	PNS	Mean follow-up of 17 (range 4–36 months) months; 80% patients with >50% reduction in monthly frequency, in the first three month of placement
Hord et al. [80]	Retrospective	Migraine	Vagus	62	PNS	25% patients with complete pain relief, 25% patients with mild pain relief, and 50% patients with a lot of pain relief; 100% patients with a dcrease in migraine frequency
Goadsby et al. [81]	Open-Label, Single-Arm, Multiple Attack, Pilot	Migraine	Vagus	30	PNS	For the first attack, after two hours of treatment, 21% patients with baseline moderate or severe headache reported pain-free and 47% patients reported partial pain relief
Kinfe et al. [82]	Prospective, Observational Cohort	Migraine	Vagus	20	PNS	Median pain VAS reduction from 8 (range 7.5 to 8) to 4 (range 3.5 to 5); mean number of headache days per month decreased by 39.5% in tota; migraine attacks per month decreased by 38.4%l
Barbanti et al. [83]	Open-Label, Single-Arm, Multicenter	Migraine	Vagus	50	PNS	56.3% patients with >50% pain reduction at one hour, 62.9% of whom reported pain-free; 64.6% patients with >50% pain reduction at two hours, 61.3% of whom reported being pain-free
Tassorelli et al. [84]	Randomized Controlled Trial	Migraine	Vagus	248	PNS	At 120 min, statistically significant difference in pain-free responder rate in nVNS compared to sham; nVNS superior to sham in abortion of the first treated attack at 30 and 60 min, but not at 120 min
Diener et al. [85]	Randomized Controlled Trial (prospective, multicenter, double-blind, parallel-group)	Migraine	Vagus	477	PNS	No statistically significant difference in the number of migraine days per month, percentage of patients with at least 50% reduction in number of migraine days, mean reductions in the number of headache days, nor reductions in the number of acute medication days
Najib et al. [86]	Randomized Controlled Trial	Migraine	Vagus	336	PNS	After 12-week of double-blind period, nVNS group with statistically significant difference in the proportion of >50% reduction in the number of migraine days
Silberstein [87]	Prospective, Multicenter, Double-Blind, Sham-Controlled, Pilot	Migraine	Vagus	73	PNS	No statistically significant difference in the number of headache days during the randomized phase; statistically significant decrease in mean headache days after eight months in the nVNS group during the open-label phase
[88]	Randomized Controlled Trial (Monocentric, Prospective, Double-Blind)	Migraine	Vagus	46	PNS	36.4% reduction in headache days in the 1 Hz group and 17.4% reduction in the 25 Hz group; 29.4% in the 1 Hz and 13.6% in the 25 Hz group reported >50% improvement in headache days
**Cervical Spinal Cord**						
**Author**	**Study Type**	**Condition**	**Targeted Nerve(s)**	**Number of Participants**	**Type of** **Intervention**	**Outcome**
Al-Kaisy et al. [90]	Prospective, Long-Term, Single-Center, Open-Label	Migraine	Cervical Spinal Cord	20	SCS (10 kHz)	At 52-week follow-up, 60% patients with >30% reduction in mean monthly migraine days (MMD) and 50% patients with >50% reduction in mean MMD
Wolter et al. [91]	Prospective	Cluster Headache	Cervical Spinal Cord	7	SCS (high cervical epidural electrodes)	Mean follow-up of 23 (range three to 78 months) months; decrease in attacks per day from 5.0 (range 1.7–10.0 attacks/day) to 1.4 (range 0–3.5 attacks/day); reduction in mean VAS from 7.4 to 4.5
Dario et al. [92]	Case	Post-Traumatic Headache	Cervical Spinal Cord	1	SCS (cephalad end of C3–C4)	80% improvement of neck pain and resolution of supraorbital pain
Zhao et al. [93]	Case	Postherpetic Neuralgia	Cervical Spinal Cord	1	SCS (C1–C2; 40 Hz)	Reduction in NRS score from 7–9 to 1–2 post-implantation
**Infraorbital Nerve**						
**Author [Citation]**	**Study Type**	**Condition**	**Targeted Nerve(s)**	**Number of Participants**	**Type of** **Intervention**	**Outcome**
Mammis et al. [95]	Case	Cluster Headache	Infraorbital	1	PNS	At 36-month follow-up, reduction in frequency of headache from three-to-four episodes per day to three-to-four episodes per month
**Great Auricular Nerve**						
**Author**	**Study Type**	**Condition**	**Targeted Nerve(s)**	**Number of Participants**	**Type of** **Intervention**	**Outcome**
Elahi et al. [96]	Case	Migraine	Great Auricular	1	PNS	At six-month follow-up, significant alleviation of pain
Elahi et al. [97]	Case	Post- Traumatic Headache	Great Auricular	1	PNS (C2-C3 branches)	At six-month follow-up, >90% reduction in headache frequency
**Auriculotemporal Nerve**						
**Author**	**Study Type**	**Condition**	**Targeted Nerve(s)**	**Number of Participants**	**Type of** **Intervention**	**Outcome**
Simopoulos et al. [100]	Case	Migraine	Auriculotemporal (bilateral)	1	PNS	At 16-month follow-up, MIDAS decreased from grade IV (severe disability) to grade II (mild disability), and average pain intensity from 8–9 to 5 on a 0–10 scale
Zhou et al. [101]	Prospective	Chronic Headache	Auriculotemporal	20	PNS	At three-month follow-up, reduction in mean total pain index (TPI) from 516 to 74.8 post-implantation
**Sphenopalatine Ganglion**						
**Author**	**Study Type**	**Condition**	**Targeted Nerve(s)**	**Number of Participants**	**Type of Intervention**	**Outcome**
Tepper et al. [103]	Prospective	Migraine	Sphenopalatine Ganglion	11	PNS	Two patients with a complete resolution of migraine headaches within three minutes of stimulation, three patients with reduction in pain, and five patients with no response, and one not stimulated
Jürgens et al. [104]	Open-Label Prospective	Cluster Headache	Sphenopalatine Ganglion	33	PNS	45% patients with >50% attacks, >50% reduction in attack frequency
Goadsby et al. [105]	Double-Blind, Randomized Controlled Trial	Cluster Headache	Sphenopalatine Ganglion	93	PNS	62.46% of SPG stimulation group with pain relief, while 38.87% of control group with pain relief
Elahi et al. [106]	Case	Facial Pain	Sphenopalatine Ganglion (pteyrgopalatine fossa)	1	PNS	Complete discontinuation of opioid-based medication

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
