# Peer review of "Neuromodulation for Craniofacial Pain and Headaches"

_biomedicines, 2023, doi:10.3390/biomedicines11123328_

Round 1

Reviewer 1 Report

Comments and Suggestions for Authors

The authors have investigated the diseases of patients affected by intractable headache,

The results are not represented in the table . I think that the authors need to report the all results in Table or histogram for easy description.

Author Response

We have inserted a summary table into the text of our paper. We appreciate the recommendation.

Reviewer 2 Report

Comments and Suggestions for Authors

In their manuscript titled “Neuromodulation for Craniofacial Pain and Headaches”, the Authors present a review that summarizes the results from the literature (Pubmed) about the use of neuromodulation for the treatment of facial pain and headaches. This is a very interesting and tamily study. It seems also well written in most of its parts.

However, what it is not totally clear to me is the study design that the Authors decided for their work. Indeed, I think that just summarizing the results from Pubmed is not sufficient, in terms of the quality and quantity of the studies retrived, neither to provide a complete overview of the issue the Authors addressed nor to give the Authors the chance to defend a point of view or a perspective or treatment indications.

In my opinion, a systematic review and, eventually, a meta-analysis could better cover the Authors’s research needs. Therefore, and this is a major point to be fixed before re-submit the manuscript to the Journal, I suggest Authors to completely redesign their work following the PRISMA guidelines for systematic review (and meta-analysis).

Author Response

Thank you for the suggestion for further work on our topic. The purpose of this paper is to summarize the current state of the literature on this topic. While we wholly agree that a full systematic review and meta-analysis would be a great next step, this is not the intended scope of our paper. We appreciate the thoughtful reply.

Round 2

Reviewer 1 Report

Comments and Suggestions for Authors

The article can be accepted